# Medullary Thyroid Carcinoma Without Calcitonin: A Case Linking Ultimobranchial Bodies to Tumor Evolution

**DOI:** 10.3390/pathophysiology32040056

**Published:** 2025-10-23

**Authors:** Ion Prisneac, Abigail I. Wald, Chelsea Bragg, John A. Ozolek

**Affiliations:** 1Department of Pathology, Anatomy and Laboratory Medicine, Robert C. Byrd Health Sciences Center, West Virginia University, Morgantown, WV 26506, USA; ionprisneac@gmail.com (I.P.); chelsea.bragg@wvumedicine.org (C.B.); 2Division of Molecular Genomic Pathology, Room 8034, Clinical Lab Building, 8th Floor, 3477 Euler Way, Pittsburgh, PA 15213, USA; waldai@upmc.edu

**Keywords:** medullary thyroid cancer, calcitonin-negative, C-cells, p63, ultimobranchial bodies

## Abstract

Medullary thyroid carcinoma (MTC) is a thyroid tumor with neuroendocrine properties purportedly derived from C-cells. The biochemical activity of medullary thyroid carcinoma includes the production of calcitonin and carcinoembryonic antigen, which are sensitive tumor markers, facilitating diagnosis, follow-up, and prognostication. Calcitonin-negative medullary thyroid carcinoma is a rare, poorly understood primary neuroendocrine carcinoma of the thyroid characterized by classic medullary thyroid carcinoma morphology without raised serum calcitonin and with or without the expression of calcitonin detected by immunohistochemistry. Previous studies reported that C-cells were derived from the neural crest; however, more recently, C-cells have been indisputably shown to be derived from the pharyngeal endoderm and ultimobranchial bodies. Ultimobranchial body (UBB) remnants can persist in the thyroid and express p63, but their function is poorly understood. Some have postulated that ultimobranchial bodies may be the “stem” cell of the thyroid and may be precursors for thyroid tumors, particularly mixed tumors with follicular and medullary components. We present a unique case of calcitonin-negative MTC in a 58-year-old male arising in an inflamed and fibrotic thyroid with numerous scattered ultimobranchial body remnants and concomitant C-cell hyperplasia/medullary microcarcinoma (CCH/MMC). The ultimobranchial body remnants, C-cell hyperplasia, and medullary thyroid carcinoma were MTC classifier positive according to ThyroSeq^®^. The areas representing CCH/MMC expressed calcitonin by IHC while the main MTC tumor was negative. An additional unique feature was an area demonstrating a “mixed” C-cell/thyroid follicular epithelial phenotype. In this review we review the possible etiologies of calcitonin-negative MTC, the possibility of a neoplastic sequential progression from ultimobranchial bodies to CCH/MMC to medullary thyroid carcinoma with the individual elements (UBB, CCH/MMC, MTC) demonstrated in this thyroid, and previous postulations that ultimobranchial bodies may be the source of some follicular thyroid cancers, medullary thyroid cancers, and mixed tumors of medullary and follicular epithelial types.

## 1. Introduction

Medullary thyroid carcinoma (MTC) is a thyroid neoplasm with neuroendocrine properties originating presumably from the neoplastic transformation of thyroid parafollicular or C-cells. C-cells were first identified by Baber in 1876 in dog thyroid and termed parenchymatous cells [1]. The ontogeny of C-cells from the cranial neural crest was concluded predominantly based on studies by LeDouarin and Pearse in the late 1960s and early 1970s [2,3,4]. However, relatively recent and unequivocal evidence shows C-cells are derived from ultimobranchial bodies and the pharyngeal endoderm [5,6,7]. They function as “neuroendocrine” cells in calcium homeostasis by production and the secretion of calcitonin (CT).

MTCs represent from 1 to 10% of all thyroid cancers; either sporadically (75% of cases) or familial (25%) because of RET proto-oncogene germline mutation. The familial form arises in the context of multiple endocrine neoplasia type 2 (MEN2A and MEN2B). Sporadic MTC is usually well differentiated and generally displays locally aggressive behavior. Familial MTC forms, especially MEN 2B, have a worse prognosis with earlier lymph node metastasis and adjacent tissue invasion [8,9]. The biochemical activity of MTC includes the production of CT and carcinoembryogenic antigen (CEA), which are sensitive tumor markers related to tumor size and facilitate diagnosis, follow-up, and prognostication for MTC [10]. However, rare MTCs occur in patients without elevation of serum calcitonin levels (calcitonin-negative MTC). These tumors may or may not express CT by immunohistochemistry or harbor RET mutations. In these cases, the diagnosis of MTC is based on expression of CEA and association with C-cell hyperplasia [11,12,13,14,15].

While C-cells are the putative origin for MTC, the link back to ultimobranchial bodies (UBB) is less certain. More recent evidence, however, shows not only the origin of C-cells from the pharyngeal endoderm but also that UBB (derived from pharyngeal endoderm) are the originating source of C-cells [5,6,7]. In addition, based on older studies, solid cell nests (SCNs) have been characterized morphologically, immunophenotypically, and ultrastructurally and determined to be UBB remnants [16,17,18]. UBB remnants have been postulated to be the cell of origin for other thyroid tumors in addition to MTC including squamous cell carcinoma, mucoepidermoid carcinoma, and mixed medullary and follicular thyroid carcinoma [19,20,21,22,23].

We present an unusual case of calcitonin-negative MTC arising in a “burnt out” thyroid where the only remaining significant cellular elements included putative UBB remnants, C-cell hyperplasia/medullary microcarcinomas (CCH/MMC), and MTCs, which were all MTC classifier positive according to ThyroSeq^®^. The CCH/MMC expressed calcitonin while it was noted the MTC did not, according to immunohistochemistry. In addition, two adjacent nodules demonstrated a mixed follicular epithelial and C-cell immunophenotype. We review the embryology of the UBB and previous literature postulating that UBB (or other pharyngeal endodermal precursors) may be the cell of origin participating in neoplastic progression from UBB to CCH/MMC to MTC and may be the origin for some follicular epithelial and mixed tumors of follicular epithelial/medullary cells.

## 2. Results

A 58-year-old male with a vague history of hypothyroidism and Hashimoto thyroiditis presented with a thyroid nodule. At the time of presentation, he was not being treated for hypothyroidism. Ultrasound with Doppler demonstrated a right thyroid midlobe single, solid, lobular, hyperemic, markedly hypoechoic nodule measuring 1.9 × 1.7 × 1.2 cm with prominent internal blood flow. Thyroid iodine uptake (I-123) was markedly depressed. The patient underwent fine-needle aspiration of the suspicious nodule with a subsequent diagnosis of “suspicious for follicular neoplasm” (Bethesda IV). Afirma^®^ was positive for MTC classifier, consistent with medullary thyroid carcinoma. Laboratory data included TSH (0.2 uIU/mL, reference 0.34–3.0), normal calcitonin level (4.0 pg/mL, reference range < 14.4), normal CEA level (2.2 ng/mL, reference 0–4.7), normal intact PTH, normal urine catecholamines, and plasma metanephrine. Calcium, phosphorus, and magnesium were also within normal ranges. Molecular testing for germline RET mutations did not reveal germline RET mutations. Based on the poorly functioning thyroid with evidence of MTC, a total thyroidectomy was performed.

The total thyroidectomy specimen was a 6.0 g diminutive thyroid with a 2.3 cm encapsulated nodule spanning the right superior and mid lobe portions. The specimen was submitted entirely and histologically showed very scant normal thyroid parenchyma and a densely fibrotic stroma with lymphoepithelial islands (LEIs) scattered throughout (Figure 1A). These lymphoepithelial islands demonstrated glandular and morular structures composed of cells containing moderate amounts of amphophilic cytoplasm and nuclei that exhibited some variation in size and shape from round to ovoid, with nuclear membrane irregularities, visible nucleoli, centrally cleared chromatin, and nuclear grooves (Figure 1B). The glandular configurations also demonstrated glomeruloid morphology with a rim of outer cells around a central sheet-like core. The dense lymphocytic component surrounded and interdigitated between epithelial cells (Figure 1B). Other areas demonstrated more identifiable squamous morules (Figure 1C) and rare cysts. The areas showing C-cell hyperplasia/medullary microcarcinoma (CCH/MMC) generally showed nodules varying in size and exhibiting predominantly follicular architecture with some follicles containing intraluminal dense pink material (Figure 1D). The cells contained a moderate amount of clear to lightly amphophilic cytoplasm with round nuclei having powdery to finely clumped “neuroendocrine” chromatin and indistinct nucleoli (Figure 1E). The MTC was a circumscribed nodule showing a few dissecting fibrous bands and microfollicular architecture with closed to slightly visible lumens and no intraluminal secretions (Figure 1F). In comparison to the nodules of hyperplastic C-cells, the cells of the MTC had moderate cytoplasm with slightly increased amphophilia, large nuclei with membrane irregularities and nuclei with visible nucleoli and more clumped chromatin (Figure 1G). Several lymphoepithelial islands resided at the periphery of the MTC (Figure 1H) that will be described below. Scant relatively normal thyroid parenchyma was seen (Figure 1I).

Table 1 and Figure 2 demonstrate the immunohistochemical expression of the epithelial elements delineated above. Notably, the epithelium (glandular, morular, glomeruloid, and cystic morphologies) within lymphoepithelial islands was exclusively highlighted by expression of galectin-3 and p63. Not all cellular constituents in these LEI expressed p63. In many instances, p63-positive cells appeared to rim or partially rim clusters of p63-negative cells. Galectin-3 expression appeared to be more widespread within the epithelial elements within the LEI but varied in intensity (Figure 2A). P63 was differentially expressed in the LEI epithelium containing the ultimobranchial remnants (Figure 2B). Epithelial types (UBB, C-cell hyperplasia/, MTC, and thyroid follicular epithelium) expressed cytokeratins 7 and 19 (Figure 2C). Calcitonin expression was limited to the nodular islands representing CCH/MMC and was negative except for very rare cells within the MTC nodule (Figure 2D,G). Otherwise, the areas of CCH/MMC and MTC displayed similar intensity and distribution of expression for IHC biomarkers, including classic neuroendocrine markers (CD56, chromogranin, synaptophysin, PGP9.5). Expression of carcinoembryonic antigen (CEA-M) was restricted to CCH/MMC and MTC (Figure 2E,F). Carbonic anhydrase IX was expressed diffusely in the MTC and only weakly in CCH/MMC (Figure 2H). CDH1/E-cadherin was strongly expressed in the CCH/MMC while the MTC showed weak to negative expression (Figure 2I). PAX8 expression was strongly expressed in the thyroid follicular epithelium, weakly and focally within the LEI, and was negative in CCH/MMC and MTC (not shown).

An interesting finding in this case was the IHC expression pattern of two adjacent nodules that differed histologically but that demonstrated expression of calcitonin, NKX2-1/TTF-1 (not shown), and neuroendocrine markers characteristic of C-cells (Table 1 and Figure 3). The nodule on the right side had a more follicular pattern with intraluminal dense eosinophilic secretions while the nodule on the left had histological characteristics more in keeping with C-cell hyperplasia (Figure 3A). Both nodules expressed CEA and calcitonin (Figure 3B,C), and neuroendocrine markers with differential expression of CK7 (Figure 3D) and CK19, thyroglobulin (Figure 3E), and PAX8 (Figure 3F).

ThyroSeq^®^ testing was performed on targeted regions representing the major epithelial components of the thyroid (excluding areas with normal morphology). The targets are shown in Appendix A and the results are given in Appendix A. All targets (UBB, CCH, CCH/MMC, MTC) demonstrated increased expression (percentage of RNA sequencing reads) of MTC classifier genes CALCA (calcitonin-related polypeptide alpha) and CHGA (chromogranin A) with the highest number of reads corresponding to the areas of CCH/MMC and MTC. The areas representing presumed UBB were also classified as MTC positive with a lower percentage of RNA sequencing reads for CALCA and CHGA genes. No mutations of the RET gene were identified.

## 3. Materials and Methods

### 3.1. Immunohistochemistry

Paraffin tissue blocks were prepared from formalin-fixed tissue processing in routine fashion. Sections at 5 µm thick were prepared for all routine hematoxylin and eosin staining and immunohistochemistry. The list of primary antibodies used and protocol details for immunohistochemistry are detailed in Table 2. All immunohistochemistry was performed on Ventana BenchMark ULTRA (Roche Diagnostics USA, Indianapolis, IN, USA) or Dako Omnis (Agilent Dako, Santa Clara, CA, USA) automated stainers.

### 3.2. Molecular

ThyroSeq testing was performed on the microdissected targets shown in Appendix A according to the methodology outlined in [24].

## 4. Discussion and Review

We present a case of calcitonin-negative medullary thyroid carcinoma, a previously described but rare entity. This calcitonin-negative MTC is unique in that it presented in the context of a fibrotic thyroid with only scant “normal” thyroid remaining with accompanying lymphoepithelial islands harboring presumably UBB remnants and nodules representing CCH/MMC. The CCH/MMC did express calcitonin while the main tumor was devoid of calcitonin expression by IHC. In addition, adjacent nodules in one section demonstrated immunophenotypes of mixed follicular and medullary epithelial elements. This raises questions again as to the mechanism of loss of calcitonin expression by these calcitonin-negative MTCs. This case also raises interesting questions regarding tumor progression and the allows us to revisit older postulates of the common source of mixed follicular and medullary tumors in the thyroid.

### 4.1. Ultimobranchial Body and C-Cell Development

As background to this possibility, the pharyngeal organs, including the thyroid gland, ultimobranchial body, thymus, and parathyroid glands, all originate from the pharyngeal endoderm, in the form of either pharyngeal pouches or thyroid diverticulum [7]. Experiments by Le Douarin with an avian chimera system demonstrated that quail neural crest transplanted to chick embryos was detected in the UBB and some of these cells seemed to expressed calcitonin. It was concluded that these migratory neural crest cells were the cells of origin for C-cells. Three papers by Le Douarin and Pearse [2,3,4] all convincingly touted the neural crest as the origin of mammalian C-cells. These publications along with the demonstration that C-cells exhibit neuroendocrine properties (part of the amine precursor uptake and decarboxylation (APUD); a term coined by Pearse) appeared to confirm that C-cells were definitively of neural crest origin [2,4,25]. However, proof that the neural crest migrates to the UBB or primordial pouch in the mammalian embryo, not the avian, provided some doubt as to the true origin of C-cells. Johansson et al. in 2015 [5], in their seminal paper, definitively demonstrated using reporter transgenic mice for specific transcription markers of both the neural crest (Wnt1) and anterior pharyngeal endoderm (Sox17) that Wnt1 expressing cells were restricted to the ectomesenchyme (neural crest) while Sox17 expression was clearly restricted to the thyroid bud, UBB, and calcitonin-expressing C-cells (endoderm). They further demonstrated that calcitonin-producing cells were derived from the UBB once integrated into the thyroid [5].

Embryologically, the UBB is composed of two types of cells: one expressing T/ebp/Nkx2.1/NKX2-1/TTF-1 and the other expressing p63. T/ebp is not required for the formation and migration of the UBB but is essential for its survival [26]. Interestingly, p63 is not essential for thyroid or UBB development but is useful as a marker of UBB and its remnants, the solid cell nest (SCN) [27]. To our knowledge, no other cell type in the thyroid expresses p63 and its continued expression and functional role in the UBB remnants remains an enigma (more on this later).

SCNs, the UBB remnants, have fascinated pathologists since they were initially described by Getzowa in 1907 [3]. SCNs are small (less than 2 mm) and are seen in roughly 3% of thyroid glands examined microscopically, predominantly in the middle and upper third of lateral lobes. Histologically, SCNs can appear as a sheet of compact blue cells in a lobular and festooning pattern mimicking a lymphoid aggregate under very low-magnification light microscopy. The individual cells can exhibit moderately high nuclear-to-cytoplasmic ratios and have slightly oval, hyperchromatic nuclei with smooth chromatin and occasional nuclear grooves. Squamous morules can be seen and indeed, at times, a squamoid appearance to the nests is the predominant morphology. In our case, a range of morphologies for the p63-positive UBB were appreciated. The predominant one was observed in the lymphoepithelial islands (LEIs) and demonstrated a glandular morphology with overlapping ovoid nuclei very closely resembling the nuclei characterizing papillary thyroid carcinoma (PTC). Rare squamous morular, cystic, and glomeruloid morphologies, all of which have been described in SCN, were also seen. Immunophenotypically, these elements demonstrated strong cellular cytoplasmic expression of cytokeratins 7 and 19, galectin-3, and nuclear expression for NKX2-1/TTF-1 and p63 consistent with published reports [16,17,18,22,23,26,28]. Notably, these cells were negative for chromogranin, calcitonin, synaptophysin, and thyroglobulin but variably and weakly positive for Bcl-2, CD56, and PGP9.5. The expression of galectin-3 and p63 distinguished these presumed UBB remnants from the other constituents of this thyroid. While one could have entertained these elements as papillary thyroid carcinoma, the lack of a mass-forming lesion, intermingling with morular and cystic structures, and lack of HBME-1 staining distinguished these as UBB remnants. It must be noted that Thyroseq testing demonstrated these cells were MTC classifier-positive, showing increased percentage of RNA sequencing reads for CALCA and CHGA genes yet negative for expression of calcitonin and chromogranin, respectively, according to IHC. The fact that these epithelial types expressed genes associated with C-cells is not altogether surprising and further supports C-cell and MTC derivation from the UBB/UBB remnants.

Indeed, p63-positive PTC has been reported [29,30] and, in our experience, we have come across several mass-forming tumors that met diagnostic criteria for PTC that demonstrated numerous p63-positive tumor cells. As noted, UBB/UBB remnants have been postulated as the possible cell of origin for some thyroid tumors including follicular epithelial derived tumors, mixed follicular-medullary tumors, squamous cell carcinomas (p63 positive), and mucoepidermoid carcinomas [19,20,22]. P63 is a member of the p53 tumor-suppressor gene family and is expressed in the basal/stem cells of several types of epithelia, including skin, esophagus, and urethra, as well as secretory epithelial tissues, including lacrimal glands, mammary glands, and prostate glands [31,32,33]. The expression of p63, Bcl-2, and telomerase has prompted speculation that the UBB may be the thyroid “stem cell” and given the possible link to other thyroid tumors, this may not be such an ontological/pathogenetic stretch [28].

In our case, the UBB remnants were represented predominantly within a lymphocytic inflammatory milieu and distributed as “islands” throughout the thyroid gland. While this inflammatory element was not scrutinized for the purposes of this presentation, the association is curious. The increased presence of SCN in association with inflammation and inflammatory thyroid diseases has been noted [34,35] and in our anecdotal experience, p63-positive cells are more readily identified on routine histological examination in heavily inflamed thyroids within lymphoid aggregates.

### 4.2. C-Cell Hyperplasia/Medullary Microcarcinoma

In addition to the UBB remnants and calcitonin-negative MTC, CCH was prominent. The CCH/MMC was predominantly in the form of small nodules present both in close association with the LEIs and separate larger nodules, hence the designation as CCH/MMC. Islands representing CCH were also seen at the periphery of the main MTC tumor mass. In all sections examined, CCH and possible MMC expressed calcitonin according to IHC and was the only epithelial component expressing calcitonin. Otherwise, CCH/MMC mirrored the immunophenotype of the MTC tumor. The cells composing the CCH/MMC were cytomorphologically like normal C-cells, having abundant clear to slightly amphophilic cytoplasm and round nuclei with a “salt and pepper” chromatin.

Normal C-cells are difficult to identify in routine H&E stain sections and are low in number (approximately 50 per low-magnification field). They are usually more centrally located within the lobes, excluding the extremes of poles and isthmus, and typically occur singly or in clusters around follicles of up to four cells. The cells are argyrophilic and identified by IHC using commercially available antibodies to calcitonin. In addition, normal C-cells express low-molecular-weight cytokeratins [8] and generic neuroendocrine markers, including chromogranin A and synaptophysin [6,9]. CCH is a multifocal proliferative condition characterized by an increased mass of C-cells within the follicles of the thyroid gland. It must be noted, however, that a precise definition of what constitutes CCH has always had some degree of latitude. CCH has been delineated as primary (neoplastic) and nonheritable CCH (secondary or physiologic CCH). Microscopically, CCH is defined by the presence of at least 50 C-cells per low-power microscopic field (100×) and may be described as either focal (partial involvement of follicles), diffuse (rimming follicles), or nodular (proliferation obliterating follicles). In addition, “primary” CCH associated with heritable MTC has sometimes been referred to as “neoplastic” CCH or thyroid intraepithelial neoplasia of C-cells (THINC). Some have distinguished neoplastic from reactive CCH by lack of staining with CD56 in reactive CCH and the necessity for cytologic atypia in neoplastic CCH. Also, to round out the spectrum, medullary microcarcinoma (MMC) has arbitrarily been defined as nodular neoplastic C-cell hyperplasia less than 1 cm in its greatest dimension [9,36]. Multiple studies have demonstrated that MTC associated with CCH/MMC represent familial disease in 98% of cases. The probability of sporadic MTC with CCH/MMC in conjunction is below 2%, making our case notable. It is likely, therefore, that a small subset of sporadic MTC is preceded by CCH/MMC that is morphologically indistinguishable from MEN 2-associated (“neoplastic”) CCH/MMC. Accordingly, molecular studies should be performed to rule out RET germline mutations in equivocal cases.

Our case demonstrated several nodular areas of CCH/MMC and small tumorlets representing MMC by current criterion. In our case, the CCH/MMC and MTC demonstrated identical IHC profiles, expressing cytokeratins 7 and 19, CEA-M, and neuroendocrine markers CD56 (neural cell adhesion molecule, also expressed in the normal thyroid), chromogranin, synaptophysin, and PGP9.5 (protein gene product 9.5, neuroblastic and neural marker) with the notable exception of lack of calcitonin expression in the MTC. Both the CCH/MMC and MTC also expressed Bcl-2, carbonic anhydrase 9 (CA-IX), and variably and weakly galectin-3 (not present in the MTC). In keeping with the molecular ontogeny of C-cells, both CCH/MMC and MTC expressed NKX2-1/TTF-1 but not PAX8. In keeping with what is known about CD56 expression in CCH/MMC and MTC, in our case, the CCH/MMC, MTC, and thyroid showed strong expression of CD56 while the UBB remnants showed weak to no expression. In contrast to our findings in this case, galectin-3 expression has been reported in MTC, but is reportedly absent from cases of C-cell hyperplasia [37]. Bcl-2 is reported to be variably expressed in advanced MTC while MEN-2 associated C-cell hyperplasia and small medullary thyroid carcinoma are strongly positive [38]. Our case also demonstrated strong expression of Bcl-2 in the CCH/MMC and MTC. There is some suggestion that downregulation of Bcl-2 may identify a subset of tumors with more aggressive clinical course [39].

### 4.3. Medullary Thyroid Carcinoma

In their review of calcitonin-negative MTC published in 2019, Gambardella et al. identified 49 cases from the literature in their exhaustive search concluding in February 2018 [12]. Since that publication, one other larger series of 19 patients from a single center in Korea was reported by Kim et al. [13]. Taking these two series together (68 tumors), 35 MTCs expressed calcitonin by IHC, 20 were negative, and 13 tumors did not have calcitonin IHC results. Likewise, 27 cases expressed CEA by IHC, 25 were negative, and 16 did not have results. RET mutations were detected in 12, not detected in 18, and not reported for 38. Of cases where both calcitonin IHC and RET mutations were available, 6/23 were positive for both calcitonin by IHC and harbored a RET mutation. Conversely, 2/23 cases were negative for both calcitonin expression and RET mutation. In cases where the histomorphology is at least consistent with MTC and calcitonin is negative by IHC, it is incumbent upon the pathologist to exclude other neoplasms that can mimic MTC. This list includes intrathyroidal parathyroid, paragangliomas, Ewing sarcoma/primitive neuroectodermal tumor, thymic carcinomas with neuroendocrine differentiation, hyalinizing trabecular tumor, and metastatic neuroendocrine tumors. Livolsi makes the excellent observation that MTC, by definition, must produce calcitonin [8,14]. The diagnosis of “calcitionin-free” or “calcitonin-negative” MTC histologically should only be made in a familial setting or if the tumor occurs in the context of C-cell hyperplasia. While 30–75% of MTCs express NKX2-1/TTF-1, CEA expression specifically supplants other entities in the differential diagnosis even in the absence of calcitonin expression [14]. In our case, the MTC expressed both NKX2-1/TTF-1 and CEA, did not express thyroglobulin, and had concomitant C-cell hyperplasia excluding other lesions. Multiple possible explanations have been proposed for the lack of calcitonin secretion and expression in calcitonin-negative MTC. These include the “hook” effect where high levels of an analyte produce a falsely low result and different calcitonins are not all recognized by the same antibody. Some have proposed that due to alternative splicing of the CT gene-related peptide (CGRP), the ratios of proCT and matureCT are altered and some undifferentiated MTCs may lose the ability to produce CT through mutations in the CT/CGRP gene [11,12,15]. As noted, the MTC in this case had the highest percentage of RNA sequencing reads for the CALCI and CHGA genes and yet did not express calcitonin according to IHC.

### 4.4. CDH1/E-Cadherin and Carbonic Anhydrase IX Expression in MTC

Of interest in our case was the expression of carbonic anhydrase IX (CA-IX) and virtual lack of expression of CDH1/E-cadherin in the MTC compared to the other epithelial elements. CA-IX and CA-XII are transmembrane proteins upregulated by hypoxic microenvironments in many tumors. CA-IX is a direct transcriptional target of hypoxia inducing factor (HIF) and plays a role in both pH regulation and cell migration, allowing survival and movement of tumor cells in a hypoxic microenvironment. In the realm of thyroid cancers, CA-XII is expressed in most thyroid tumor types while CA-IX is relegated to expression in MTC and anaplastic thyroid tumors. Differentiated thyroid follicular tumors (follicular carcinomas and papillary thyroid carcinomas) show virtually no expression of CA-IX. Evidence suggests a role for RET-mediated activation of the HIF pathway leading to increased CA-IX expression in MTC [40]. The MTC in our case strongly expressed CA-IX while the presumed UBB remnants and C-cell hyperplasia showed no to weak expression according to IHC, perhaps giving credence to a RET-mediated activation despite no evidence of RET mutation according to ThyroSeq.

As noted, CDH1/E-cadherin expression by IHC was markedly reduced in our MTC compared to the UBB remnants, CCH/MMC, and normal thyroid. In keeping with our observations here, down-regulation of CDH1/E-cadherin in MTC is characteristic of the invasive phenotype and is simultaneous with transient loss of Foxa2 expression in malignant C-cells during epithelial-to-mesenchymal transition [5,41]. Foxa2 is essential for the formation and patterning of pharyngeal endoderm. Foxa2 expression is seen in the thyroid bud and pharyngeal pouch but is downregulated in the follicular epithelium at the bud stage. Downregulation of Fox2a is necessary for delamination of the UBB, at which time both Fox1a and Foxa2 are then re-expressed in a differential pattern. Proximal UBB cells (still connected to the pharyngeal pouch) demonstrated proliferation by Ki-67 labelling and Foxa1 positive cells and distal UBB cells were Ki-67 negative and expressed Foxa2. UBB cells disperse peripherally throughout the thyroid and maintain expression of Foxa1, Foxa2, and calcitonin distinguishing themselves from thyroglobulin positive follicular cells. MTC and RET-mutant MTC cell lines both show high levels of FOXA1 and FOXA2 transcripts. FOXA1 expression appears to promote growth of MTC cells and Foxa2 downregulation is associated with invasive character [5,20,41].

### 4.5. UBBs: The Cell of Origin for Thyroid Cancers?

A primary reason for reporting this case was to show both morphologically and genetically that a relationship may exist pathogenetically between the UBB, CCH/MMC, and MTC and that this case may indeed demonstrate a neoplastic sequence of progression from the UBB to cancer of C-cells. It was noted earlier that some of the epithelial elements, particularly within LEIs, that were marked as UBB had glandular profiles and nuclear features almost indistinguishable from those seen in papillary thyroid carcinomas. One could legitimately question whether these epithelial elements within the LEI described in this case were indeed UBB and that is legitimate, with the caveat that as far as we know, UBB/UBB remnants are the only epithelial elements that express p63 in the thyroid. In addition, characteristic squamous morules heralded as constituents of the SCN (UBB remnants) accompanied these glandular and glomeruloid profiles. Speculation that MTC may represent a neoplastic sequence originating with the UBB is not novel. This has been postulated as the ontogeny of so-called mixed medullary and follicular thyroid carcinomas (MMFTCs) where the follicular epithelium is intermixed with neoplastic C-cells (not collision tumors) [20]. Parenthetically, one must question whether this concept should be restricted to just tumors where the neoplastic follicular epithelium is intermingled with the neoplastic C-cell component. Zhang et al. in their recent series of synchronous medullary and papillary thyroid carcinomas demonstrated that tumors described as Type III and Type IV (tumors separated by normal thyroid parenchyma within the same lobe or different lobes) occurred four times more frequently than intermingled or collision tumors (Types I and II, respectively) [42]. The concept of a single cell of origin for such rare tumors makes more sense teleologically if UBBs are indeed of endodermal embryological phylogeny. Otherwise, mechanisms must be postulated and investigated to show how cells of neuroectodermal (neural crest derived) origin can attain features and phenotypes of differentiated follicular epithelium. UBBs, as noted, have molecular features of putative stem cells; so, multiphenotypic differentiation is not entirely implausible. Some experimental evidence does point to a possible common cell of origin for MMFTCs, including bilateral mixed tumors with a germline RET mutation and occurrence of mixed tumors in children and mixed tumors in cattle where the fusion of the UBB and thyroid is not complete [42].

Lastly but importantly, the differential expression of IHC markers in adjacent nodules is perplexing but of serious interest and may lend some credence to the hypothesis that MTC (at least in this case) is a neoplastic progression of UBB remnants demonstrating some phenotypic plasticity for both C-cell and follicular epithelial types. On HE, both nodules had similar cytomorphology. The left nodule demonstrated morphology and immunophenotyping of CCH/MMC and MTC; notably, lack of PAX8 and thyroglobulin expression and expression of CEA and neuroendocrine markers. Puzzling, however, was the lack of expression of cytokeratins 7 and 19, both of which were unequivocally expressed in the UBB remnants, CCH/MMC, and MTC. The right nodule, even more so, had a mixed follicular and C-cell phenotype displaying expression of thyroid follicular epithelial markers (PAX8, cytokeratin, thyroglobulin) and all neuroendocrine markers (synaptophysin, chromogranin A, CD56, PGP9.5) as well as CEA. A clear molecular embryological explanation for the IHC findings in these adjacent nodules is not apparent based on what is known regarding thyroid, UBB, and C-cell development and should not be speculated to any great extent in this case.

## 5. Summary

We present a case of calcitonin-negative medullary thyroid carcinoma against a background of extensive C-cell hyperplasia/medullary thyroid microcarcinomas and numerous lymphoepithelial islands containing ultimobranchial body remnants in a background of a “burnt out” thyroid. All epithelial elements were MTC classifier-positive according to Thyroseq. This thyroid also contained a focal area where two adjacent nodules demonstrated mixed thyroid follicular epithelial and C-cell elements. These findings lend some support to the suggestion that the UBB, C-cells, and MTC all derive from the same cell type and may lend credence to the postulation that the UBB may be the stem cell of the thyroid and the cell of origin for tumors of C-cells, some thyroid epithelial tumors, and mixed thyroid follicular epithelial and medullary tumors. This notion is further strengthened from clear evidence that the thyroid follicular epithelium, UBB, and C-cells are derived from the pharyngeal endoderm.

## Figures and Tables

**Figure 1 pathophysiology-32-00056-f001:**
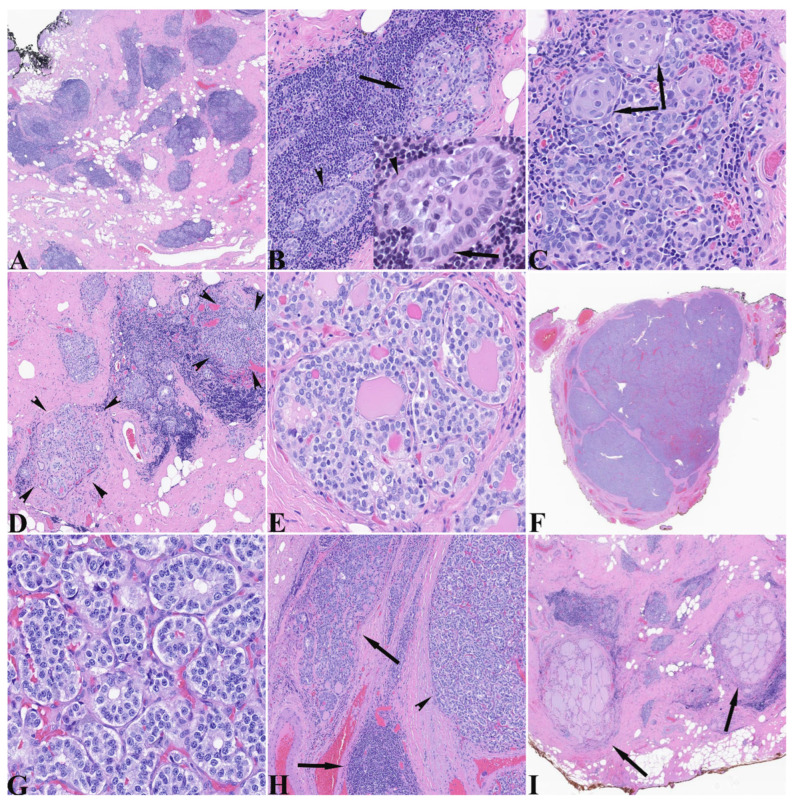
(**A**) Low magnification representative view of section of thyroid showing numerous blue “islands” amidst a fibrofatty stroma with no apparent normal thyroid follicular parenchyma (HE, 20×). (**B**) Representative “lymphepithelial island” showing epithelial elements in a background of small lymphocytes. The epithelium demonstrates glandular microarchitecture (arrow) and structures composed of a single layer of epithelium rimming a central epithelial core (arrowhead). Inset shows high magnification view of structure identified by the arrowhead. The rimming layer is composed of cells with round to elliptical nuclei that overlap. Occasional nuclei show nuclear grooves (inset arrow) and central clearing (inset arrowhead) (HE, 600×). (**C**) In addition to the above epithelial microarchitectures, some epithelium had a squamoid nested pattern (squamous morules) (arrows) (HE, 400×). (**D**) Small nodules composed of microfollicles (arrowheads), some with dense eosinophilic intraluminal material, were present. Note the intervening lymphepithelial island (HE, 40×). (**E**) Higher magnification view of one of the microfollicular nodules shows cells having rather uniform round nuclei, some with small visible nucleoli and generally smooth chromatin. The dense eosinophilic intraluminal material is readily apparent (HE, 400×). (**F**) The large nodule representing the medullary thyroid carcinoma has a circumscribed appearance with some traversing thin fibrous bands conferring a multinodular pattern (HE, 5×). (**G**) In contrast to the cells composing the CCH/MMC, the MTC cells had larger nuclei, more prominent nucleoli, and a “salt and pepper” chromatin. Note the lack of intraluminal material (HE, 400×). (**H**) Distinctive lymphoepithelial islands and foci demonstrating microfollicles with intraluminal material (arrows) were immediately adjacent to the MTC (arrowhead) (HE, 100×). (**I**) Scant “normal” appearing thyroid parenchyma was seen in this diminutive thyroid (arrows) (HE, 20×).

**Figure 2 pathophysiology-32-00056-f002:**
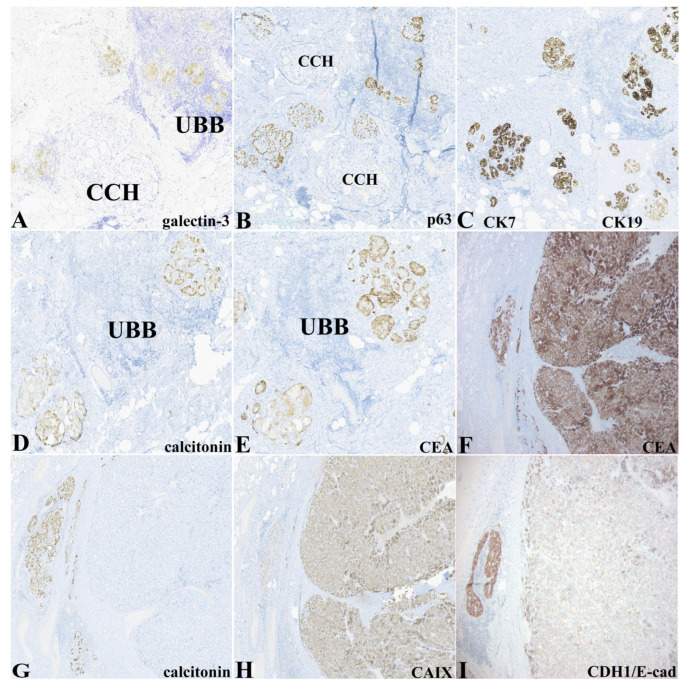
(**A**) Galectin-3 expression within epithelial elements of lymphoepithelial islands (LEI) (upper right). The adjacent small focus of C-cell hyperplasia is negative (lower left) (galectin-3, 100×). (**B**) P63 expression is restricted to the epithelium within the LEI while the C-cell nodules are negative (p63, 100×). (**C**) Both the C-cell nodules and epithelium within the LEI strongly express cytokeratin 7 and cytokeratin 19 (inset) (CK7, 100× and CK19, 100×). (**D**) Conversely, calcitonin expression is restricted to the C-cell hyperplasia/medullary microcarcinoma foci (CCH/MMC) (calcitonin, 100×). (**E**) CCH/MMC expressed CEA uniformly while negative in LEI (CEA, 100×). (**F**) MTC (large right nodule in G-L) and surrounding compressed islands resembling CCH diffusely expressed carcinoembryonic antigen (CEA, 40×). (**G**) Calcitonin expression in a compressed island of microfollicles at the periphery of the medullary thyroid carcinoma (MTC) tumor while the main tumor mass is negative for calcitonin (calcitionin, 50×). (**H**) The MTC showed diffuse expression of carbonic anhydrase IX while the peripheral nodule (left) was negative) (CAIX, 50×). (**I**) Lack of CDH1/E-cadherin expression in the MTC (right) while the compressed island at the periphery (CCH/MMC) demonstrated strong diffuse expression (CDH1/E-cadherin, 100×).

**Figure 3 pathophysiology-32-00056-f003:**
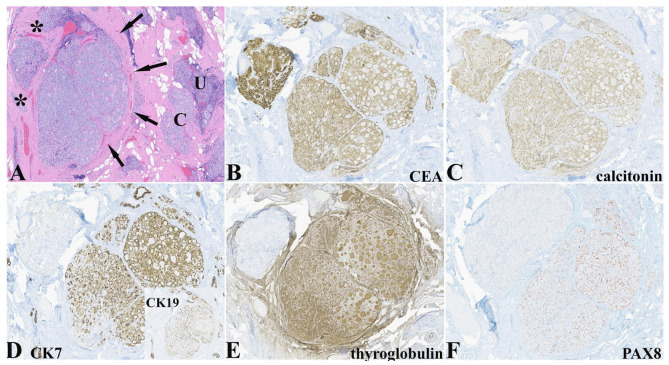
(**A**) One focus in this thyroid resection revealed two adjacent nodules (left bounded by asterisks and right by arrows). These nodules were next to a lymphoepithelial island (LEI) (“U”) and C-cell hyperplasia/medullary microcarcinoma (CCH/MMC) (“C”) (HE, 50×). (**B**) Both nodules expressed CEA in similar fashion (CEA, 50×). (**C**) Both nodules expressed calcitonin in a diffuse and similar fashion (calcitonin, 50×). (**D**) The right nodules demonstrated differential expression of cytokeratins 7 and 19 (inset) compared to no expression of these in the left nodule (cytokeratin 7, 50× and cytokeratin 19, 50×). (**E**) The right nodule showed expression of thyroglobulin diffusely while the left was negative (thyroglobulin, 50×). (**F**) The right lower nodule expresses PAX8 while the left upper is negative (PAX8, 50×).

**Table 1 pathophysiology-32-00056-t001:** Immunohistochemical expression of biomarkers in epithelial components: LEI—lymphoepithelial islands, UBBs—ultimobranchial bodies, CCH/MMC—C-cell hyperplasia/medullary microcarcinoma, MTC—medullary thyroid carcinoma, L/R—Left upper nodule (Figure 3) in left column and right lower nodule in right column, NP-IHC not performed.

Antibody	LEI/UBB	CCH/MMC	MTC	Thyroid	Adjacent Nodules (L/R)	
**Bcl-2**	+	++	+++	+++	+++	+++
CA-IX	−/+	+	+++	−	NP	NP
Calcitonin	−	++	−	−	++/+++	++/+++
CD56	+	+++	+++	+++	+++	+++
CEAM	−	++	+++	−	+++	+++
Chromogranin	−	+++	+++	−	NP	NP
CK7	+++	++	++	+++	−	+++
CK19	+++	++	+++	+++	−	+++
CDH1/E-cadherin	+++	+++	−/+	+++	+++	+++
Galectin-3	+++	−/+	−	−	−	−
HBME-1	−/+	−	−	−	NP	NP
p63	+++	−	−	−	−	−
PAX8	+	−	−	+++	−	+++
PGP9.5	+	+++	+++	−	+++	+++
Synaptophysin	−	+++	+++	−	+++	+++
TTF1	+++	+++	+++	+++	+++	+++
Thyroglobulin	−	−	−	+++	−	+++

“+” represents extent of expression, “–” no expression. + low extent, ++ moderate extent, +++ all of lesion stained.

**Table 2 pathophysiology-32-00056-t002:** Specifications of antibodies used for immunohistochemistry; CEA—carcinoembryonic antigen.

Antibody	Species	Clone	Vendor	Platform
Calcitonin	rabbit monoclonal	SP17	Cell Marque for Ventana	BenchMark ULTRA
CD56	rabbit	MRQ-42	Ventana	BenchMark ULTRA
CEA	mouse	CEA31	Cell Marque for Ventana	BenchMark ULTRA
Chromogranin A	mouse	LK2H10	Ventana	BenchMark ULTRA
CK AE1/AE3	mouse	AE1/AE3	Cell Marque	BenchMark ULTRA
Cytokeratin 7	rabbit monoclonal	SP52	Ventana	BenchMark ULTRA
Cytokeratin 19	mouse monoclonal	A53-B/A2.26	Cell Marque	BenchMark ULTRA
CDH1/E-cadherin	mouse monoclonal	36	Ventana	BenchMark ULTRA
Galectin-3	mouse	9C4	Cell Marque	Dako Omnis
HBME1	mouse	HBME-1	Cell Marque	Dako Omnis
Ki67	rabbit monoclonal	30-9	Ventana	Dako Omnis
P63	mouse	4A4	Ventana	BenchMark ULTRA
PAX-8	mouse	MRQ-50	Cell Marque for Ventana	BenchMark ULTRA
PGP 9.5	rabbit	Polyclonal	Cell Marque	Dako Omnis
Synaptophysin	rabbit monoclonal	SP11	Ventana	BenchMark ULTRA
Thyroglobulin (Quest)	mouse monoclonal		Beckman Coulter	BenchMark ULTRA
NKX2-1/TTF-1	rabbit monoclonal	SP141	Ventana	BenchMark ULTRA

## Data Availability

No additional datasets were analyzed or generated by this study.

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
