# Peer review of "Medullary Thyroid Carcinoma Without Calcitonin: A Case Linking Ultimobranchial Bodies to Tumor Evolution"

_pathophysiology, 2025, doi:10.3390/pathophysiology32040056_

Round 1
Reviewer 1 Report
Comments and Suggestions for Authors
Based on a rare case of medullary thyroid carcinoma (MTC), the authors provide histopathological and immunocytochemical findings of tumor heterogeneity representing different tumor foci that display dissimilar expression pattern of known biomarkers. That all individual tumors (n=4) display neuroendocrine features consistent with MTC was corroborated by ThyroSeq analysis. One of the tumors in addition expressed thyroglobulin suggesting a possible mixed medullary-follicular tumor phenotype. Tumor cell origins are discussed with reference to thyroid development and the mixed population of cells in the ultimobranchical bodies (UBB) from which thyroid C-cells derive.
This is an interesting case report illustrating the occasional complex features and diagnosis difficulties of thyroid neoplasms. There are some issues that should be revised and suggestions of improvement (in order of appearance):
- Abstract. The abbreviation of medullary thyroid carcinoma (MTC) is given but not consequently used. Please check manuscript throughout.
- Introduction, page 2 second paragrah, please add refs to statement of endoderm origin of thyroid C cells (cited below but should be be included already here).
- Case, page 3. This section should rather read ´Results´. Please adhere order of presentation to journal´s format (Introduction, Materials and Methods, Results, Discussion).
- Page, 3, first paragraph, case report details should preferably also include whether the patient was on substitution therapy due to hypothyroidism, dates of diagnosis and surgery, whether there were any signs of tumor spreading regional of systemic (if not, for how long disease-free post-surgery).
- Page 3, second paragraph, please rephrase to ´... right superior and mid lobe portions´.
- Page 3, second paragraph further below, it seems reference to Fig. 1F is missing.
- Page 4, end of first paragraph, for PAX8 expression please refer to the appropriate figure(s).
- Page 4, second paragraph, TTF-1 should be replaced by the formal name NKX2-1, and TTF-1 with explanation of the abbreviation provided as additional information. Since TTF-1 probably is the established name among clinical pathogists, NKX2-1/TTF-1 might thereafter be used throughout for improved clarity including in Tables. Moreover, with reference to legend pf Fig. 3, this figure panel does not contain any NKX2-1/TTF-1 IHC image.
- Page 4, third paragraph, ThyroSeq data referring to Table 3 should include not only expression profiles but also findings of the mutation analysis. It is stated that none of the examined tumors were RET mutation positive, but did you not find any mutations or genomic aberrations (e.g. LOH)?
- Fig. 2 (page 6). Please add biomarker identity to each IHC image as this will certainly improve readability of the panel.
- Figs. 2 and 3 (page 6 and 7). High power images missing now are required to be able to distinguish positively and negatively IHC-stained cells, in particular this is important to provide for the mixed MTC/follicular carcinoma tumor.
- Page 7, end of first paragraph, the formal name for E-cadherin to be used is CDH1. You may use CDH1/E-cadherin in the following including Tables.
- Page 10, second paragraph, in the discussion of P63+ cells it is relevant to mention squamous metaplasia in PTC and squamous cell carcinomas of the thyroid recently classified as anaplastic thyroid cancer.
- Page 12, last paragraph, please forward the cited paper to ´... during epithelial-to-mesenchymal transition (13)´.
Reviewer 2 Report
Comments and Suggestions for Authors
The manuscript presents an interesting case of calcitonin-negative medullary thyroid carcinoma (MTC), with unique pathological features that support the hypothesis of a shared pharyngeal endoderm origin for ultimobranchial bodies (UBB), C-cells, and MTC. The study is well-structured, and supported by appropriate charts and diagrams. However, the following suggestions are offered to enhance the manuscript:
- While the authors cite two papers (Gambardella et al., 2019; Kim et al., 2021) in Discussion Section 4.3 to compare pathological features, a more in-depth discussion of the diagnostic, therapeutic, and prognostic distinctions between calcitonin-negative MTC and classic MTC would be valuable. For instance:
How does the absence of calcitonin expression affect diagnostic sensitivity and specificity, particularly in preoperative settings?
Are there differences in treatment response or long-term outcomes for calcitonin-negative MTC compared to classic MTC?
Could the molecular mechanisms underlying calcitonin negativity (e.g., alternative splicing of CT/CGRP or RET-independent pathways) influence therapeutic strategies?
- In the case report, the patient has a history of hypothyroidism and Hashimoto’s thyroiditis, with a TSH level below the reference range. To provide a clearer clinical context, the following details are suggested to be included:
Is the patient on thyroid hormone replacement therapy? If so, the dosage and duration should be noted.
Levels of free T3 (FT3), free T4 (FT4), and thyroid autoantibodies (e.g., TPOAb, TgAb) would help clarify the thyroid functional status and autoimmune activity.
- Given that 2–6% of MTC patients develop ectopic ACTH production leading to Cushing’s syndrome (Ilias I, Torpy DJ, Pacak K, Mullen N, Wesley RA, Nieman LK. Cushing's syndrome due to ectopic corticotropin secretion: twenty years' experience at the National Institutes of Health. J Clin Endocrinol Metab. 2005;90(8):4955-4962. doi:10.1210/jc.2004-2527). It would be pertinent to: state the ACTH level in this case to rule out or confirm this association.
- Minor Editorial Suggestions:
Abbreviations: In the abstract, the first mention of abbreviations (e.g., UBB) should include the full term (e.g., "ultimobranchial bodies (UBB)").
Figure Legends: Ensure consistency in labeling (e.g., "Figure 2A" vs. "2D" should be standardized to "Figure 2A, 2D" or similar).
Reviewer 3 Report
Comments and Suggestions for Authors
Dear Authors,
I would like to sincerely thank you for your interesting and thought-provoking case report titled "Medullary Thyroid Carcinoma Without Calcitonin: A Case Linking Ultimobranchial Bodies to Tumor Evolution." The topic you addressed is both rare and of growing relevance in the field of thyroid pathology, and I truly appreciated the clarity of your description and the depth of the discussion.
Please find below a few suggestions that I hope may contribute to further enhancing your valuable work:
I would recommend replacing the keyword “medullary thyroid carcinoma” with either “MTC” or “medullary thyroid cancer”, to better align with commonly used search terms and terminology.
It would be helpful to include more clinical details about the lesion: its location within the thyroid, size, ultrasound features, and whether associated lymphadenopathies were present. A brief table format could also be considered to improve clarity.
I was also curious about the rationale behind the choice of total thyroidectomy, especially considering this was a solitary nodule.
Could you kindly clarify whether the patient had any family history of thyroid or endocrine neoplasms?
Lastly, I would suggest harmonising the references in accordance with the journal’s formatting requirements and referring to the Instructions for Authors for Pathophysiology, which recommend including a final "Conclusion" section in case reports. In this regard, you might consider moving the current summary to the beginning of the manuscript.
Thank you again for your contribution to this complex and fascinating topic. I look forward to reading more from your research group in the future.
Round 2
Reviewer 1 Report
Comments and Suggestions for Authors
Comments and suggestions of improvement have been satisfactory considered and revised.
Reviewer 3 Report
Comments and Suggestions for Authors
I would like to thank the authors for providing clarification on this fundamental aspect. By including these informations the manuscript may be considered suitable for
publication.